# Symptomatic and Asymptomatic Patients in the Polish Atrial Fibrillation (POL-AF) Registry

**DOI:** 10.3390/jcm10051091

**Published:** 2021-03-05

**Authors:** Marek Kiliszek, Beata Uziębło-Życzkowska, Iwona Gorczyca, Małgorzata Maciorowska, Olga Jelonek, Beata Wożakowska-Kapłon, Maciej Wójcik, Robert Błaszczyk, Monika Gawałko, Agnieszka Kapłon-Cieślicka, Tomasz Tokarek, Renata Rajtar-Salwa, Jacek Bil, Michał Wojewódzki, Anna Szpotowicz, Małgorzata Krzciuk, Janusz Bednarski, Elwira Bakuła-Ostalska, Anna Tomaszuk-Kazberuk, Anna Szyszkowska, Marcin Wełnicki, Artur Mamcarz, Paweł Krzesiński

**Affiliations:** 1Department of Cardiology and Internal Diseases, Military Institute of Medicine, 04-141 Warsaw, Poland; buzieblo-zyczkowska@wim.mil.pl (B.U.-Ż.); mmaciorowska@wim.mil.pl (M.M.); pkrzesinski@wim.mil.pl (P.K.); 21st Clinic of Cardiology and Electrotherapy, Swietokrzyskie Cardiology Centre, Collegium Medicum, The Jan Kochanowski University, 25-369 Kielce, Poland; iwona.gorczyca@interia.pl (I.G.); olga_jelonek@wp.pl (O.J.); bw.kaplon@poczta.onet.pl (B.W.-K.); 3Department of Cardiology, Medical University of Lublin, 20-059 Lublin, Poland; m.wojcik@umlub.pl (M.W.); robertblaszczyk1@wp.pl (R.B.); 41st Department of Cardiology, Medical University of Warsaw, 02-097 Warsaw, Poland; mgawalko@wum.edu.pl (M.G.); agnieszka.kaplon@gmail.com (A.K.-C.); 5Department of Cardiology and Cardiovascular Interventions, University Hospital, 30-688 Krakow, Poland; tomek.tokarek@gmail.com (T.T.); rajfura@op.pl (R.R.-S.); 6Department of Intensive Care and Perioperative Medicine, Jagiellonian University Medical College, 30-688 Kraków, Poland; 7Department of Invasive Cardiology, Centre of Postgraduate Medical Education, 02-507 Warsaw, Poland; biljacek@gmail.com (J.B.); michaljerzywojewodzki@gmail.com (M.W.); 8Department of Cardiology, Regional Hospital, 27-400 Ostrowiec Świętokrzyski, Poland; szpotowiczanna@wp.pl (A.S.); kardiologia@zoz.ostrowiec.pl (M.K.); 9Department of Cardiology, St. John Paul II Western Hospital, 05-825 Grodzisk Mazowiecki, Poland; medbed@wp.pl (J.B.); elwira.bakula@gmail.com (E.B.-O.); 10Department of Cardiology, Medical University of Bialystok, 15-276 Białystok, Poland; a.tomaszuk@poczta.fm; 11Department of Cardiology, University Hospital of Bialystok, 15-276 Białystok, Poland; anna.szyszkowska92@gmail.com; 123rd Department of Internal Diseases and Cardiology, Warsaw Medical University, 02-091 Warsaw, Poland; artur.mamcarz@mssw.pl (M.W.); 3klinika@mssw.pl (A.M.)

**Keywords:** atrial fibrillation, EHRA class, registry

## Abstract

Background: Atrial fibrillation (AF) can cause severe symptoms, but it is frequently asymptomatic. We aimed to compare the clinical features of patients with asymptomatic and symptomatic AF. Methods: A prospective, observational, multicenter study was performed (the Polish Atrial Fibrillation (POL-AF) registry). Consecutive hospitalized AF patients over 18 years of age were enrolled at ten centers. The data were collected for two weeks during each month of 2019. Results: A total of 2785 patients were analyzed, of whom 1360 were asymptomatic (48.8%). Asymptomatic patients were more frequently observed to have coronary artery disease (57.5% vs. 49.1%, *p* < 0.0001), heart failure with preserved ejection fraction (39.8% vs. 26.5%, *p* < 0.0001), a previous thromboembolic event (18.2% vs. 13.1%, *p* = 0.0002), and paroxysmal AF (52.3% vs. 45.2%, *p* = 0.0002). In multivariate analysis, history of electrical cardioversion, paroxysmal AF, heart failure, coronary artery disease, previous thromboembolic event, and higher left ventricular ejection fraction were predictors of a lack of AF symptoms. First-diagnosed AF was a predictor of AF symptoms. Conclusions: In comparison to symptomatic patients, more of those hospitalized with asymptomatic AF had been previously diagnosed with this arrhythmia and other cardiovascular diseases. However, they presented with better left ventricular function and were more frequently treated with cardiovascular medicines.

## 1. Introduction

Atrial fibrillation (AF) is a supraventricular arrhythmia that substantially influences patients’ morbidity and mortality. AF can be asymptomatic, or it can cause diverse symptoms of varying severity that influence patients’ quality of life. The European Heart Rhythm Association (EHRA) has proposed a score for symptoms of AF, according to which asymptomatic patients are categorized into class I, mildly symptomatic patients are categorized into class IIa and IIb, highly symptomatic patients are categorized into class III, and patients with disabling symptoms are categorized into class IV [1].

AF increases the risk of thromboembolic complications, especially stroke [2]. In asymptomatic patients, diagnosis of AF may be delayed, and the first symptom of AF may be a thromboembolic incident. There is a significant amount of evidence that numerous patients with a stroke of unknown origin have AF [3].

Several attempts have been made to describe asymptomatic AF patients, but there are still significant discrepancies in the number of such patients that have been reported and their clinical characteristics [4,5,6,7,8]. The aim of this study is to compare the clinical presentation of patients with asymptomatic and symptomatic AF.

## 2. Materials and Methods

### 2.1. Patients

The analysis was carried out on a group of patients from the Polish Atrial Fibrillation (POL-AF) registry. The methodology has been previously described in detail [9,10]. The POL-AF registry is a prospective, observational, multicenter study. Consecutive AF patients were enrolled at ten cardiology centers (ClinicalTrials.gov: NCT04419012). The data were collected in 2019 (January to December) for two weeks each month. The registry included all consecutive patients with AF, regardless of the reason for hospitalization (excluding ablation). The inclusion criteria were an age of at least 18 years and a history of documented AF. The exclusion criterion was admittance to a hospital to undergo AF ablation (it was acknowledged that patients undergoing AF ablation have a clinical profile different from most patients with AF, as they are younger and have fewer concomitant diseases). The flowchart of the analysis is shown in Figure 1.

The investigators collected copious data, including demographics, type of AF, medical history and concomitant diseases, diagnostic test results, and pharmacotherapy. The estimated glomerular filtration rate (eGFR) was calculated using the Chronic Kidney Disease Epidemiology Collaboration (CKD-EPI) equation [11]. CHA2DS2-VASc [12] and HAS-BLED [13] scores were calculated according to contemporary guidelines [1].

The Ethics Committee of the Swietokrzyska Medical Chamber in Kielce approved the study (104/2018), and waived the requirement of obtaining informed consent from the patients.

### 2.2. Statistical Analysis

The Shapiro–Wilk test was used to test the normality of the distribution of continuous variables. Continuous variables were presented as means (standard deviation, SD) or medians (interquartile range (IQR): 1st–3rd quartile). Categorical variables were presented as numbers and frequencies. Comparisons of the asymptomatic and symptomatic patients in terms of the continuous variables were performed with a Student’s t-test (normal distribution) or Mann–Whitney U test (non-normal distribution, ordinal data). A chi-square test with Yates’s correction was used for categorical variables. Two-tailed tests were performed with the Bonferroni correction, assuming about 50 simultaneous comparisons, so the *p*-value was considered significant at 0.001 and lower.

A multivariable logistic regression analysis was performed with parameters that were statistically significant in direct comparisons (asymptomatic vs. symptomatic patients) if data were available for more than 80% of cases. For this analysis, the level of significance was set at 0.05. Statistical calculations were performed with Statistica v. 12 (Statsoft Inc., Tulsa, OK, USA).

## 3. Results

Data were collected from the 2785 patients regarding their AF symptoms and EHRA class. In total, 1360 patients were asymptomatic (48.8%), while 467 patients (16.7%) had severe symptoms (EHRA classes III and IV) (Figure 1).

The general characteristics of the study group are shown in Table 1, Table 2 and Table 3.

More patients classified as EHRA I, compared with EHRA class ≥ II, had coronary artery disease (CAD) and heart failure, but only heart failure with preserved ejection fraction (HFpEF). Heart failure with reduced ejection fraction was slightly more frequent in patients with EHRA ≥II. In EHRA I, paroxysmal AF was more frequent, history of atrial fibrillation was about one year longer, and history of electrical cardioversion was more frequent. In addition, these patients had a significantly higher rate of previous thromboembolic events. Among the 211 patients with first-diagnosed AF, only 60 (28.4%) were asymptomatic. Patients with symptoms related to AF were more frequently hospitalized due to heart failure and had higher total cholesterol levels, but they had a slightly lower left ventricular ejection fraction, smaller left atrium (LA) dimension, and smaller left ventricular diastolic dimension.

CHA2DS2-VASc and HAS-BLED scores did not significantly differ between groups, although asymptomatic patients more frequently had CHA2DS2-VASc scores of 3 or higher. The frequency of oral anticoagulant treatment did not differ between groups. Symptomatic patients were less commonly treated with angiotensin-converting enzyme (ACE) inhibitors/sartans, aldosterone antagonists, and statins. Symptomatic patients were more often treated with amiodarone, but less commonly treated with class I antiarrhythmic drugs (Table 4).

According to the results of the multivariate analysis, history of cardioversion, paroxysmal AF, heart failure, CAD, previous thromboembolic event, and higher left ventricular ejection fraction were independently predictive of asymptomatic AF. First-diagnosed AF was an independent predictor of AF symptoms (Table 5).

## 4. Discussion

We showed that about half of the patients with any history of AF who were admitted to hospitals were asymptomatic. Asymptomatic patients were more likely than symptomatic ones to have CAD, heart failure, paroxysmal AF, history of cardioversion, previous thromboembolic events, and higher left ventricular ejection fraction. In contrast, symptomatic patients were more likely to have first-diagnosed AF.

Symptoms are a very important part of clinical assessments of patients with AF, and they influence the strategy of treatment, pharmacotherapy, and invasive treatment [1]. They are not associated with risk of death or major cardiovascular events, but they significantly increase the risk of unplanned hospitalization [6,7]. AF symptoms assessed with the help of EHRA class well correlated with the specific Atrial Fibrillation Effect on the Quality-of-Life (AFEQT) questionnaire [7]. Based on long-term monitoring, only about 24% of patients have symptomatic AF episodes [14]. Many more (42%) have only asymptomatic AF episodes, and 32% have both symptomatic and asymptomatic episodes [15].

With the development of novel methods of screening and diagnosing arrhythmias, asymptomatic AF—also called silent or subclinical AF—is a growing problem [16,17]. It is not clear how to classify and treat such patients. Even characterization of this group of AF patients is problematic because of significant heterogeneity of the studies [8]. Currently, it is not recommended to include subclinical AF as a part of AF classification [17].

Our results suggest that among hospitalized patients, almost half of those with AF are asymptomatic. This indicates that the rate of asymptomatic AF may be higher than previously thought. A previous study with a large cohort of ambulatory AF patients reported a frequency of 38% [7]. Very similar results were reported by the Atrial Fibrillation (EORP-AF) Pilot General Registry [4]. However, other researchers have reported significantly fewer asymptomatic AF patients (12–13%) [5,6]. Interestingly, after one-year follow-up in the EORP-AF Registry, 23% of symptomatic patients were reported, without significant differences between paroxysmal, persistent, and permanent AF [18].

It is not clear why asymptomatic patients more frequently had paroxysmal AF in our population. In the majority of other publications, nonparoxysmal AF was a factor linked with asymptomatic AF [4,5,6,7]. However, it is possible that our results were due to the population we analyzed. We included hospitalized patients, almost two-thirds of whom were not hospitalized due to AF. Additionally, the prevalence of CAD in comparison with symptomatic patients was higher among asymptomatic patients in our study. The results of other studies are not consistent. In the AFFIRM study, the prevalence of CAD was lower in the asymptomatic group [6], while in the EORP-AF Registry, previous myocardial infarction was more frequent among asymptomatic patients [4]. In a study performed by Freeman et al., the rate of CAD was comparable between the symptomatic and asymptomatic groups [7].

Male gender in numerous studies was more prevalent in the asymptomatic AF group in comparison with the symptomatic AF group [4,5,6,7,8]. In our cohort, there was no relationship between AF symptoms and gender.

More thromboembolic events could be expected in asymptomatic patients; the time from diagnosis in that group is one year longer than in symptomatic patients, and it is likely that the time between the onset of arrhythmia and diagnosis is longer. Similar observations have been reported for the AFFIRM population [6].

We found that among hospitalized patients, first-diagnosed AF is usually symptomatic. Diagnosis of AF in the general population is still symptom-driven, so in the majority of cases, first-diagnosed AF is symptomatic.

In our cohort, the proportion of heart failure diagnoses was relatively high (66%). In Freeman et al.’s cohort, the proportion was 32% [7], and in the EORP-AF Registry, it was 47.5% [18]. Interestingly, heart failure was more prevalent in the group of asymptomatic AF patients, but only HFpEF. Reduced left ventricular ejection fraction was more prevalent in the symptomatic AF group.

The subjectivity of patients’ self-assessment could also determine how our results should be interpreted. It has been shown that almost 25% of AF patients without HF diagnoses had significantly impaired exercise capacity (VO2peak below 16 mL/kg/min in a cardiopulmonary exercise test). Self-reported exercise tolerance was poorly correlated with objective measures from cardiopulmonary exercise testing [19]. Therefore, if physicians consider only patients’ reports, they may overestimate the rate of asymptomatic patients [7].

Our results showed that symptomatic patients in comparison with asymptomatic ones were less commonly treated with ACE inhibitors/sartans, aldosterone antagonists, or statins. However, although those differences were statistically significant, they were not clinically relevant. This was probably the result of the higher prevalence of previously diagnosed cardiovascular diseases (CAD and heart failure). The same trend regarding ACE inhibitors and statins was seen in a US cohort [7]. Interestingly, in the EORP-AF Registry, asymptomatic patients were more likely to be treated with ACE inhibitors and aldosterone antagonists [4].

Symptomatic patients were more often treated with amiodarone and less commonly with class I antiarrhythmic drugs. In the Belgrade Atrial Fibrillation study, amiodarone was more commonly applied in the asymptomatic group [5]. In the AFFIRM trial, antiarrhythmic drugs were more commonly given (before randomization) to the symptomatic group [6]. It is likely that antiarrhythmic drugs made the patients asymptomatic by decreasing the number of AF episodes and shortening them.

Our results reveal the real-world characteristics of hospitalized AF patients. The identified differences between symptomatic and asymptomatic patients provoke some important questions, such as the following: How many patients are objectively symptomatic? How should we diagnose symptomatic AF and differentiate it from heart failure (e.g., symptoms, biomarkers)? In how many AF patients were heart-failure symptoms and reduced left ventricular ejection fraction transient? Is pharmacotherapy (ACE inhibitors, statins, sartans, aldosterone antagonists) the key to determining why there was a positive relation between comorbidities and asymptomatic AF? Is this a coincidence or a casual relation? Why does a substantial percentage of patients with asymptomatic AF receive antiarrhythmic drugs (14–32% in previous studies) [4,5,6,7]?

All these considerations prove that the assessment of AF patients is complex and in every case the specific clinical settings should be carefully analyzed.

### Limitations

This was a large, multicenter registry with a large number of AF patients. However, some centers did not report the patients’ EHRA class, decreasing the number of patients for which there were available data. There were missing data for some parameters (e.g., time from first AF diagnosis, echocardiography parameters, cholesterol level), which prevented the use of those parameters in the multivariate analysis. The major limitation of the study is its pure observatory character and the registry-derived data. There was no follow-up, which meant that we could only perform a comparison of baseline characteristics. In addition, we had no data on heart rate and blood pressure at the time of symptom assessment. Further, it is probable that some patients were symptomatic at the time of admission and became asymptomatic after treatment.

Given the significant heterogeneity of previous studies [8], our results provide a substantial amount of contemporary data about symptomatic and asymptomatic AF patients. With the development of novel tools of AF detection, the characteristics of asymptomatic AF patients will probably change over time.

The sample selection could have influenced the results. Symptomatic patients are more likely to be admitted to a hospital due to AF, including first-diagnosed AF. Some of them may be hemodynamically decompensated, leading to the presentation of heart-failure symptoms. However, asymptomatic patients are usually hospitalized due to other cardiovascular diseases (i.e., CAD, heart failure, hypertension), which may bias the difference in comorbidities and pharmacotherapy between the analyzed groups.

## 5. Conclusions

Almost half of the patients with a history of AF were asymptomatic. Coronary artery disease, heart failure, paroxysmal AF, history of cardioversion, and previous thromboembolic events were more common in asymptomatic AF patients, while first-diagnosed AF was more common in symptomatic patients.

## Figures and Tables

**Figure 1 jcm-10-01091-f001:**
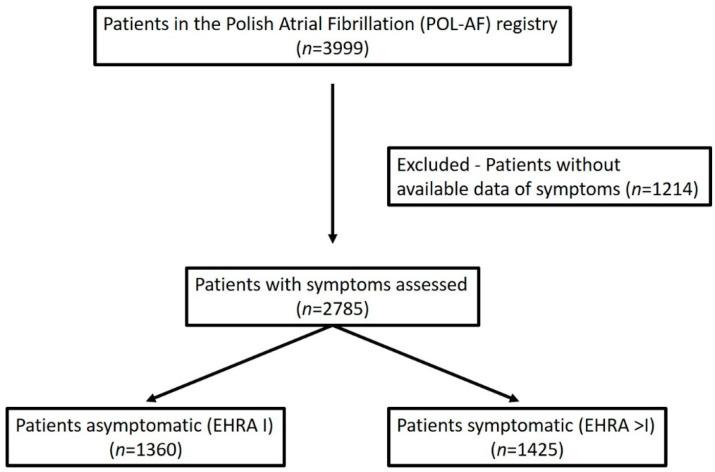
Flowchart of the analysis. EHRA—European Heart Rhythm Association score.

**Table 1 jcm-10-01091-t001:** Reasons for hospital admission and AF characteristics.

	Total	EHRA I	EHRA > I	*p*
Reasons for admission, *n* (%)				
AF	994 (35.7)	480 (35.3)	514 (36.1)	0.67
CIED implantation	233 (8.4)	122 (9.0)	111 (7.8)	0.26
Planned coronary angiography/angioplasty	343 (12.3)	181 (13.3)	162 (11.4)	0.13
Acute coronary syndrome	199 (7.1)	94 (6.9)	105 (7.4)	0.64
Heart failure	458 (16.4)	159 (11.7)	299 (21.0)	<0.0001
AF characteristics, *n* (%)				
History of cardioversion	783 (28.1)	457 (33.6)	345 (24.2)	<0.0001
AF at admission	1893 (68.0)	941 (69.2)	952 (66.8)	0.19
First-diagnosed AF	211 (7.6)	60 (4.4)	151 (10.6)	<0.0001
Paroxysmal AF	1355 (48.7)	711 (52.3)	644 (45.2)	0.0002
Time from first AF diagnosis * (years), mean(SD)	4.8 (4.7)	5.3 (4.7)	4.3 (4.6)	<0.0001

Abbreviations: AF, atrial fibrillation; CIED, cardiovascular implantable electronic device; EHRA, European Heart Rhythm Association score. * data available in < 80% of patients.

**Table 2 jcm-10-01091-t002:** Demography and concomitant diseases.

	Total	EHRA I	EHRA > I	*p*
Demographic data				
Age, mean (SD)	72 (11)	72 (11)	72 (11)	0.84
Female gender, *n* (%)	1181 (42.4)	556 (40.9)	625 (43.9)	0.12
Concomitant diseases, *n* (%)				
Hypertension	2405 (86.4)	1172 (86.2)	1233 (86.5)	0.83
Diabetes	1000 (35.9)	479 (35.2)	521 (36.6)	0.48
Heart failure	1842 (66.1)	961 (70.7)	881 (61.8)	<0.0001
HFrEF	650 (23.7)	292 (21.9)	358 (25.4)	0.033
HFmrEF	287 (10.5)	138 (10.3)	149 (10.6)	0.88
HFpEF	951 (34.7)	531 (39.8)	374 (26.5)	<0.0001
Coronary artery disease	1481 (53.2)	782 (57.5)	699 (49.1)	<0.0001
Previous myocardial infarction	662 (23.8)	311 (22.9)	351 (24.6)	0.29
Chronic kidney disease	778 (27.9)	366 (26.9)	412 (28.9)	0.26
Previous thromboembolic incident	434 (15.6)	248 (18.2)	186 (13.1)	0.0002
Previous bleeding	78 (2.8)	41 (3.0)	37 (2.6)	0.58
Thromboembolism and Bleeding Risk Scores				
CHA_2_DS_2_-VASc score (points), median (IQR)	5 (4–6)	5 (4–6)	5 (3–6)	0.10
CHA_2_DS_2_-VASc ≥ 3 (points), mean (SD)	2485 (89.2)	1249 (91.8)	1236 (86.7)	<0.0001
HAS-BLED score (points), median (IQR)	2 (2–3)	2 (2–3)	2 (2–3)	0.13

Abbreviations: HFmrEF, heart failure with mid-range ejection fraction; HFpEF, heart failure with preserved ejection fraction; HFrEF, heart failure with reduced ejection fraction; IQR, interquartile range; EHRA, European Heart Rhythm Association score.

**Table 3 jcm-10-01091-t003:** Laboratory and echocardiography findings.

	Total	EHRA I	EHRA > I	*p*
Laboratory data, mean (SD)				
Hemoglobin, g/dL	13.18 (1.96)	13.15 (0.50)	13.21 (0.50)	0.67
Creatinine, mg/dL	1.259 (0.74)	1.09 (0.9–1.38)	1.10 (0.9–1.4)	0.03
Total cholesterol, mg/dL *	162 (52)	155 (50)	168 (53)	<0.0001
Echocardiography data, median (IQR)				
LVEF, %	54 (40–60)	55 (40–60)	52 (40–60)	0.001
LA, mm *	47 (42–51)	48 (43–53)	46 (42–50)	<0.0001
LA area, cm square *	29.0 (25.0–35.0)	29.1 (25.0–34.9)	29.0 (25.0–35.0)	0.54
LVDd, mm *	52 (47–58)	53 (48–58)	51 (47–57)	<0.0001

Abbreviations: LA, left atrium antero-posterior dimension; LVDd, left ventricular end-diastolic dimension; LVEF, left ventricular ejection fraction; EHRA, European Heart Rhythm Association score. * data available in < 80% of patients.

**Table 4 jcm-10-01091-t004:** Pharmacotherapy.

Type of Treatment, *n* (%)	EHRA I	EHRA > I	*p*
OAC	1122 (82.8)	1152 (81.1)	0.27
Amiodarone	209 (15.6)	332 (23.5)	<0.0001
I class AAD	176 (13.1)	101 (7.2)	<0.0001
Beta blockers	1186 (88.4)	1212 (85.8)	0.048
ACE inhibitors/sartans	1094 (81.6)	1069 (75.7)	0.0002
Aldosteron antagonists	612 (45.6)	504 (35.7)	<0.0001
Calcium channel blockers	509 (38.0)	481 (34.1)	0.037
Statins	1077 (80.3)	1051 (74.4)	0.0003

Abbreviations: AAD, antiarrhythmic drugs; ACE, angiotensin-converting enzyme; OAC, oral anticoagulants; EHRA, European Heart Rhythm Association score.

**Table 5 jcm-10-01091-t005:** Results of multivariable logistic regression analysis—predictors of symptomatic AF.

Parameter	HR	95% CI	*p*
First-diagnosed AF	2.29	1.61–3.24	<0.001
Paroxysmal AF	0.80	0.67–0.96	0.016
History of electrical cardioversion	0.65	0.53–0.79	<0.001
Heart failure	0.59	0.48–0.72	<0.001
Coronary artery disease	0.78	0.65–0.93	0.006
Previous thromboembolic event	0.77	0.61–0.97	0.03
Left ventricular EF	0.99	0.98–0.99	0.013

Abbreviations: CI, confidence interval; HR, hazard ratio. Other abbreviations: see Table 1 and Table 3.

## Data Availability

Data available upon request.

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
