# Peer review of "Symptomatic and Asymptomatic Patients in the Polish Atrial Fibrillation (POL-AF) Registry"

_jcm, 2021, doi:10.3390/jcm10051091_

Round 1

Reviewer 1 Report

Currently, the manuscript (MS) of Kiliszek et al. is low-average quality. Nevertheless, a prospective multi-centre study is interesting for epidemiology and overall study of atrial fibrillation (AF) and I think it might be suitable for considering publication by Editors after MAJOR REVISIONS.

Summing up, among POINTs of WEAKNESSES, we can list:

  • English language must be checked (i.e. sentences such as "In comparison to symptomatic patients, more of those hospitalized with asymptomatic AF were previously diagnosed with this arrhythmia and other cardiovascular ..." should be totally re-written).
  • About METHODS (globally a point of STRENGHT of paper) I'd like to receive some elucidations:
    • regarding exclusion of patients (pts) undergoing ablation, why? PLEASE EXPLAIN and/or INSERT any opportune valid reference if emerging by similar studies
    • authors have used CKD-EPI for assessing eGFR, why? Even if very often in AF studies (for examples RCTs for direct anti-coagulants) was used Cockroft-Gault. PLEASE CLARIFY.
    • About statistics:  I'd like to read what is the significance of the sentence depending on the distribution of the parameters. ADD any useful further EXPLANATION.
  • Quality of tables may be improved for gaining them more clearness. PLEASE TRY.
  • Finally, I'd like to suggest the following readings for enlarging discussion and/or conclusions sectors:
    • Clin Cardiol. 2017;40(6):413-418. doi: 10.1002/clc.22667.
    • Int J Cardiol. 2015;191:172-7. doi: 10.1016/j.ijcard.2015.05.011.
    • J Interv Card Electrophysiol. 2020;59(3):495-507. doi: 10.1007/s10840-020-00859-y.

With my best regards.

Author Response

Reviewer 1

Currently, the manuscript (MS) of Kiliszek et al. is low-average quality. Nevertheless, a prospective multi-centre study is interesting for epidemiology and overall study of atrial fibrillation (AF) and I think it might be suitable for considering publication by Editors after MAJOR REVISIONS.

Summing up, among POINTs of WEAKNESSES, we can list:

  • English language must be checked (i.e. sentences such as "In comparison to symptomatic patients, more of those hospitalized with asymptomatic AF were previously diagnosed with this arrhythmia and other cardiovascular ..." should be totally re-written).

The manuscript underwent second proofreading by a native speaker (scribendi.com).

  • About METHODS (globally a point of STRENGHT of paper) I'd like to receive some elucidations:
    • regarding exclusion of patients (pts) undergoing ablation, why? PLEASE EXPLAIN and/or INSERT any opportune valid reference if emerging by similar studies

Explained – page 2, Material and Methods, 1st paragraph. No explicit exclusion criteria were defined to avoid biased selection of patients and achieve a cohort close to ‘real life’. Furthermore, consecutive patients were included at each site in order to reduce selection bias. Only patients admitted to hospital to have to AF ablation were excluded from the registry because not all the centres perform catheter ablation. What is more, it was acknowledged that patients undergoing ablation due to AF have a clinical profile different from most patients with AF (they are younger and have less concomitant diseases)

    • authors have used CKD-EPI for assessing eGFR, why? Even if very often in AF studies (for examples RCTs for direct anti-coagulants) was used Cockroft-Gault. PLEASE CLARIFY.

It was the only method available in all centers.

    • About statistics:  I'd like to read what is the significance of the sentence depending on the distribution of the parameters. ADD any useful further EXPLANATION.

Explained – page 2, statistical methods.

  • Quality of tables may be improved for gaining them more clearness. PLEASE TRY.

We rearranged the tables.

  • Finally, I'd like to suggest the following readings for enlarging discussion and/or conclusions sectors:
    • Clin Cardiol. 2017;40(6):413-418. doi: 10.1002/clc.22667.
    • Int J Cardiol. 2015;191:172-7. doi: 10.1016/j.ijcard.2015.05.011.
    • J Interv Card Electrophysiol. 2020;59(3):495-507. doi: 10.1007/s10840-020-00859-y.

Added in the discussion, page 6 3rd and 6th paragraph.

With my best regards.

Reviewer 2 Report

The study presented by Kiliszek et al. represents a large collective of patients and is clear in its approach.

I have to mention some formal aspects: 

Tables: Please mention what is mentioned in the brackets, are these percent?  Sometime it seems to be average values, sometimes deviation? please clarify. 

Please change Ca blockers to calcium channel blockers. 

Content: 

How do the authors justify the higher degree of asymptomatic patients on class I AAD as these data are derived from a time before the EAST study was published? 

It has to be mentioned, that the study is  limited to it pure observatory character and the registry-derived data. Which are although represented in the baseline-charactersists of large RCTs. Here the authors should line out what their study is adding? 

Author Response

Reviewer 2

The study presented by Kiliszek et al. represents a large collective of patients and is clear in its approach.

I have to mention some formal aspects: 

Tables: Please mention what is mentioned in the brackets, are these percent?  Sometime it seems to be average values, sometimes deviation? please clarify. 

Added in the tables.

Please change Ca blockers to calcium channel blockers. 

 Changed, table 4.

Content: 

How do the authors justify the higher degree of asymptomatic patients on class I AAD as these data are derived from a time before the EAST study was published? 

Added in the text, 6th page, penultimate paragraph.

It has to be mentioned, that the study is  limited to it pure observatory character and the registry-derived data. Which are although represented in the baseline-charactersists of large RCTs. Here the authors should line out what their study is adding? 

Added in the Limitation section, page 7.

Round 2

Reviewer 1 Report

I have much appreciated all modifications to the paper and I think now it is suitable for considering publication by Editors.